# Radiation Exposure and Surgical Outcomes after Antegrade Sclerotherapy for the Treatment of Varicocele in the Paediatric Population: A Single Centre Experience

**DOI:** 10.3390/jcm12030755

**Published:** 2023-01-18

**Authors:** Carolina Bebi, Marco Bilato, Dario Guido Minoli, Erika Adalgisa De Marco, Michele Gnech, Irene Paraboschi, Luca Boeri, Irene Fulgheri, Roberto Brambilla, Mauro Campoleoni, Giancarlo Albo, Emanuele Montanari, Gianantonio Manzoni, Alfredo Berrettini

**Affiliations:** 1Department of Paediatric Urology, Fondazione IRCCS Ca’ Granda Ospedale Maggiore Policlinico, 20122 Milan, Italy; 2Department of Urology, Fondazione IRCCS Ca’ Granda Ospedale Maggiore Policlinico, 20122 Milan, Italy; 3Department of Vascular Surgery, Fondazione IRCCS Ca’ Granda Ospedale Maggiore Policlinico, 20122 Milan, Italy; 4Medical Physics Unit, Fondazione IRCCS Ca’ Granda Ospedale Maggiore Policlinico, 20122 Milan, Italy; 5Department of Clinical Sciences and Community Health, University of Milan, 20122 Milan, Italy

**Keywords:** varicocele, paediatric varicocele, antegrade sclerotherapy, Tauber procedure, male infertility, radiation exposure, fluoroscopy

## Abstract

Introduction: Antegrade sclerotherapy (Tauber) effectively treats varicocele. However, fluoroscopy exposes young males to ionizing radiation. We aimed to evaluate radiation exposure and surgical outcomes after the Tauber procedure. Materials and methods: We retrospectively analysed data from 251 patients. Dose area product (DAP) and fluoroscopy time were recorded. The effective dose was calculated with the PCXMC software. Descriptive statistics and linear regression tested the association between clinical predictors and radiation exposure. Results: Median (IQR) age and body mass index (BMI) were 14 (13–16) years and 20.1 (17.9–21.6) kg/m². Five (2.1%) patients developed clinical recurrence and two (0.81%) developed complications. Median fluoroscopy time and DAP were 38.5 (27.7–54.0) s and 89.6 (62.5–143.9) cGy*cm^2^. The effective dose was 0.19 (0.14–0.31) mSv. Fluoroscopy time was higher in patients with collateral veins (41 (26–49) s vs. 36 (31–61) s, *p* = 0.02). The median amount of sclerosing agent (SA) used was 3 (3–4) ml. DAP was higher when SA > 3 mL was used (101.4 (65–183) cGy*cm^2^ vs. 80.5 (59–119) cGy*cm^2^; *p* < 0.01). At univariable linear regression, age, BMI, operative time and SA > 3 mL were associated with higher DAP (all *p* < 0.01). At multivariable linear regression, only BMI (beta 12.9, *p* < 0.001) and operative time (beta 1.9, *p* < 0.01) emerged as predictors of higher DAP, after accounting for age and SA > 3 mL. Conclusions: The Tauber procedure is safe and associated with low effective doses. Operative time and the patient’s BMI independently predict a higher radiation dose.

## 1. Introduction

Varicocele is a common condition characterised by the abnormal dilation of pampiniform plexus veins due to venous reflux. It becomes more frequent with puberty, reaching a prevalence of 10–15% among adolescents [1]. Although most commonly asymptomatic, it can interfere with testicular growth during pubertal development or it can cause discomfort/pain. Moreover, varicocele has also been linked to infertility in the adult population [2]. The exact mechanisms by which impaired blood drainage of the testis can hinder testis development and spermatogenesis are to be clarified yet. Since varicocele treatment has been correlated with testicular catch-up growth and improved sperm parameters [3,4], a variety of surgical techniques have been developed. Traditionally, varicocelectomy was performed with an open approach by ligating the internal spermatic vein at the inguinal, sub-inguinal, or supra-inguinal level. More recently, less invasive approaches have been proposed not merely to improve patient comfort, but with the aim to spare the lymphatic vessels and decrease the rates of postoperative hydrocele [5]. 

Over the years, microsurgical varicocelectomy, together with laparoscopic-assisted and even robotic approaches were described to achieve these goals [6,7,8]. Similarly, fluoroscopy-guided antegrade sclerotherapy and percutaneous retrograde embolization techniques have gained popularity due to their minimal invasiveness, protection of the testicular artery, low complication rates, reduced operative times and fast recovery rates compared to open approaches [9]. To date, none of the aforementioned procedures have shown to be superior in terms of complication and recurrence rates [5,10,11]. However, fluoroscopy-guided techniques have often been criticised for their radiation burden on young males that are submitted to surgery for benign disease [12]. The 2022 European Association of Urology (EAU) guidelines also state that radiation exposure is less controllable with the antegrade technique [13]. Noticeably, this statement is based on old studies, with small cohorts, in which the X-ray source used during surgery was probably different compared to the ones used in current days [14,15]. 

Moreover, to the best of our knowledge, no study has ever provided detailed quantitative measures of radiation doses received by the patient during the procedure in the paediatric population. Therefore, in this article, we aimed to assess and correlate surgical outcomes and radiation exposure in paediatric patients submitted to fluoroscopy-guided antegrade sclerotherapy for the treatment of varicocele. 

## 2. Materials and Methods

We performed a retrospective analysis of data collected from 251 young males affected by varicocele and treated with antegrade sclerotherapy at a single tertiary referral centre between January 2010 and December 2021. Clinical data, perioperative characteristics and surgical outcomes were recorded for all patients. Varicocele diagnosis was based on the physical examination with or without ultrasound assessment. Indications for treatment were the presence of testicular asymmetry and/or symptomatic clinical grade III varicocele. All treated varicoceles in our cohort were left-sided. 

### 2.1. Surgical Technique

All surgeries were performed under spinal, local or general anaesthesia, as per the patient and/or anaesthesiologist’s preference. A wide-spectrum antibiotic prophylaxis was administered before surgery. To perform the procedure, the patient is positioned in the supine position. A 3–4 cm incision at the scrotal root level is performed. After opening the scrotum, the pampiniform plexus veins are identified and the most dilated and straight vessel is selected, isolated and prepared for cannulation. The venous catheter employed is a 20–24 G, 7 cm long, hollow, flexible and transparent cannula with an oblique cut ending that is inserted in the selected vein. After assessing the patency of the vein by irrigating with saline, the fluoroscopic C-arm is positioned and a contrast agent is injected to allow for visualization of the spermatic vein up until its convergence with the renal vein. The sclerosing agent (*Aethoxysklerol* or *Fibrovein*) is then mixed with air to make a foam and is immediately injected into the spermatic vein under fluoroscopic vision. This allows for modulating the amount of sclerosing agent which is injected until the complete disappearance of the contrast medium is clearly seen on the C-arm monitor. The cannula is then removed carefully, avoiding any spillage around the site of the entrance by a normal saline flush. The vein is ligated cranially and caudally, and fascia, subcutaneous tissue and skin are closed with absorbable sutures. All patients underwent a routine clinical visit seven days post-operatively and were instructed to report any complication or deviation from a normal postoperative course (minimal, slowly reducing scrotal swelling was considered normal). Clinical recurrence was evaluated at least one month after surgery. Recurrence was assessed clinically by physical examination with or without ultrasound evaluation.

### 2.2. Radiation Exposure Measurements

All procedures included the use of fluoroscopy and were performed in an operating room using a C-arm with an imaging intensifier (Vision R, Ziehm Imaging, Nürnberg, Germany), always used in automatic modality, mostly with 12.5 pulses/s. The X-ray tube was placed in an undercouch position, X-ray field collimation was adjusted by the radiographer and the image intensifier was as close as possible to the patient body. 

Fluoroscopy time was recorded for each patient. We investigated two different exposure parameters, namely the Dose Area Product (DAP) and the Effective Dose (ED). DAP is expressed in centigrade per centimeter squared (cGy*cm^2^) and is defined as the incident dose multiplied by the beam area. It is typically measured by a DAP meter located between the X-ray source and the patient, usually calculated and displayed by the X-ray system and stored in the DICOM header in order to be retrieved.

The ED expressed in millisieverts (mSv) is calculated for the whole body by adding up the equivalent doses to every organ, each multiplied by a specific tissue weighing factor, therefore taking into account the different radiation sensitivities of different organs in the body. To be noticed, the ED is calculated to a reasonable approximation by means of the PCXMC software, which uses a Monte Carlo method [16]. In particular, for the calculation of each specific ED, we employed each patient’s biometrics whenever available. Instead, for the calculation of Effective Doses in patients for whom pre-operative body measurements were unavailable, we designed a model that calculates the ED based on a conversion factor derived from the application of the Monte Carlo method in patients with complete biometric data of the same age. 

The Effective Dose provides an overall estimate of radiation exposure, and its use has two main advantages. First, it represents the indicator that most directly correlates to the radiation risk to which patients are subjected. Second, ED calculations can be reported as a point of comparison for a variety of different interventional radiology procedures. Therefore, ED can be easily used to compare radiation exposures in a plethora of settings, allowing physicians to better comprehend the real impact of radiation on the patient during Tauber procedures.

Data collection followed the principles outlined in the Declaration of Helsinki. All patients signed an informed consent agreeing to share their own anonymous information for future studies. The study was approved by our hospital Ethical Committee (Prot. 2021-ESQLFDI).

### 2.3. Statistical Analysis

Data distribution was tested with the Shapiro–Wilk test. Data are presented as medians (interquartile range; IQR) or frequencies (proportions). Descriptive statistics were used to describe the whole cohort. Univariable (UVA) and multivariable (MVA) linear regression analyses tested the association between clinical predictors and measures of radiation exposure. Statistical analyses were performed using SPSS v.26 (IBM Corp., Armonk, NY, USA). All tests were two-sided and the statistical significance level was determined at *p* < 0.05. 

## 3. Results

Table 1 details the perioperative characteristics of patients submitted to antegrade sclerotherapy. 

Above all, the median (IQR) patient’s age and body mass index (BMI) were 14 (13–16) years and 20.1 (17.9–21.6) kg/m^2^, respectively. The volume of the affected testicles was reduced in 179 (71.3%) children. The median operative time was 35 (25–41) min and the fluoroscopy time was 35 (30–45) s. Collateral veins were found in 114 (45.4%) cases and sclerosing agents (SA) used were *Aethoxysklerol* and *Fibrovein* in 117 (46.6%) and 134 (53.4%) children, respectively. The median amount of SA used was 3 (3–4) mL. 

Table 2 describes post-operative complications and recurrence rates of the study cohort. Among 251 patients, 50 were lost to longer-term follow-up. Only two (0.81%) patients developed a post-operative complication, that is a single case of postoperative scrotal swelling with haematoma and a case of homolateral hydrocele that developed in a failed procedure. At a median follow-up of 6 (3–12) months, clinical recurrence was noted in five (2.1%) patients. In the whole cohort, the median fluoroscopy time and DAP were 38.5 (27.7–54.0) s and 89.6 (62.5–143.9) cGy*cm^2^, respectively, with an effective dose of 0.19 (0.14–0.31) mSv. Fluoroscopy time was higher in patients with collateral veins than those without (41 (26–49) s vs. 36 (31–61) s, *p* = 0.02). DAP was higher in cases where >3 mL of SA were used (101.4 (65–183) cGy*cm^2^ vs. 80.5 (59–119) cGy*cm^2^; *p* < 0.01). At Spearman’s correlation, the patient’s BMI (rho 0.38, *p* < 0.001) and operative time (rho 0.27, *p* < 0.01) were positively correlated with higher DAP.

Univariable linear regression analysis showed that age, BMI, operative time and >3 mL of SA used were all associated with higher DAP (all *p* < 0.01). At multivariable linear regression analysis, only BMI (beta 12.96, *p* < 0.001) and operative time (beta 1.99, *p* = 0.017) emerged as predictors of higher DAP, after accounting for age and >3 mL of SA used (Table 3).

## 4. Discussion

Varicocele is one of the treatable causes of male infertility [17]. Over the years, a variety of approaches have been proposed to correct venous reflux at the level of the pampiniform plexus [10]. However, according to the most recent literature [5,9,11], no technique has been proven to be superior with respect to the others. Antegrade sclerotherapy, as first popularised by Tauber and Johnsen in 1994 [18], has been proposed as an alternative to retrograde sclerotherapy besides laparoscopic and microsurgical procedures. According to our findings, antegrade sclerotherapy is both effective and safe for the treatment of varicocele, with a recurrence rate of only 2.1% and a complication rate of 0.81%. As a matter of fact, only one patient developed a minor complication, that is a postoperative spontaneously resolving scrotal swelling and haematoma. We also incurred in a single failed procedure, where cannulation of the spermatic vein was not optimal as documented by the reflux of sclerosing agent in the pampiniform plexus vein collaterals, for which reason the procedure was interrupted. The patient developed a homolateral hydrocele and underwent surgical correction together with a repeated successful Tauber procedure one year later.

Although rare cases of thrombus migration and pulmonary embolism are reported in the literature after varicocele embolization, we did not incur any major thromboembolic event in our cohort. Of clinical importance, our study is the first to consider and analyse radiation exposure during antegrade scrotal sclerotherapy. As a matter of fact, we recorded a median effective dose of only 0.19 mSv in our cohort. Moreover, as expected, at multivariable analysis patients’ BMI and operative time were found to be independent predictors of radiation dose. Our dose estimations gain relevance when compared to currently available data in the literature. A previous study by Malekzabeh et al. [19] reported a median effective dose of 1.01 mSv in a cohort of young males of comparable age submitted to retrograde radiological embolization, which is four times higher than the one we reported for Tauber procedures. Moreover, to make a comparison, a median effective dose of 0.19 mSv during antegrade sclerotherapy for varicocele treatment has the same order of magnitude and is therefore comparable to the one received for common diagnostic procedures, such as abdominal X-rays, according to available literature [20]. Therefore, our study challenges the preconceived idea that radiation exposure is a major limitation of antegrade sclerotherapy techniques. This likely stems from the very first papers discussing the technique, which are based on small cohorts and antiquated technology, thus unlikely to reflect the reality of current antegrade sclerotherapy for the treatment of varicocele. As a matter of fact, the 2022 EAU pediatric urology guidelines state that angiographic occlusion of the spermatic veins meets the requirements for lymphatic sparing repair, which has a strong recommendation to prevent hydrocele formation and testicular hypertrophy [13]. However, the same guidelines criticise fluoroscopy-guided techniques for their radiation burden, stating that the radiation exposure is less controllable with the antegrade approach. Noticeably, this statement is based on dated studies examining a small cohort of patients and providing little detail on the radiation burden. Thon et al. [14], in 1989, presented one the first experiences of percutaneous sclerotherapy for varicocele treatment in 31 children from 10 to 16 years of age, affirming that fluoroscopic radiation exposure could last as long as 14 min (mean 4.9 min). Fifteen years later, Wunsch et al. [21] examined a very large cohort of pediatric patients treated with retrograde sclerotherapy, the vast majority of which were treated from 1981 to 1996. The authors stated that the radiation dose depended on the duration of the probing of the internal spermatic vein with dose area product measurements at 50–150 cGy*cm^2^, by using appropriate apertures, although statistical analysis was not provided. In 2010, Fayad et al. [15] also described their experience with percutaneous retrograde endovascular occlusion (PREVO) in 71 pediatric varicocele patients, treated with high success rates and low complication rates between the years 1990 and 2004. However, no data was provided on the radiation burden of the procedure. To conclude, further studies providing a detailed analysis of radiation exposure are needed in order to confirm our findings, as currently available literature does not provide adequate information on this matter.

Additionally, the authors believe that limiting ionizing radiation exposure in embolization techniques is of paramount importance for both general and reproductive health. The risk of secondary malignancies from radiation exposure increases with increasing doses of radiation [22]. Moreover, studies on males that are occupationally exposed to low radiation doses demonstrate increased percentages of pathological sperm cells, decreased sperm motility, and intensification of vacuolization [23,24,25]. Every effort should be made to recognise radiation risk, promote radiation safety, and optimize dose reduction techniques, especially when treating young males that are affected by a benign pathology that may represent a major fertility issue when adolescents become young adults. As a matter of fact, varicocele can be found in 25.4% of men with abnormal semen parameters [2]. Varicocele, however, is a developmental condition, and early diagnosis and treatment can improve sperm parameters and promote testicular catch-up growth [3,26,27]. 

Our study is not devoid of limitations. First, it is retrospective in nature, with the limitations of its design. Second, it is a single-centre study, which raises the possibility of selection biases; therefore, further studies are needed to validate our findings.

Lastly, a significant number of patients (50/251) only underwent an immediate post-operative evaluation seven days post-op and were lost to longer-term clinical follow-up and/or did not undergo a post-operative US, which may lead to an underestimation of recurrence rates among our cohort. As already mentioned [28], however, this issue is often encountered when attempting to follow-up benign pathologies that are treated with high success rates, particularly in the adolescent population. 

## 5. Conclusions

Antegrade sclerotherapy is a safe and effective technique for the treatment of paediatric varicocele. Although often criticised for its radiation burden, our study is the first to provide an accurate evaluation of radiation exposure during Tauber procedures. According to our findings, radiation doses during the procedure are minimal and independently predicted by operative time and the patient’s BMI. Varicocele becomes increasingly common during puberty and often becomes a major fertility issue when patients become young adults. Adequate prevention is paramount and antegrade sclerotherapy should be considered an appropriate option for the treatment of varicocele in the paediatric population.

## Figures and Tables

**Table 1 jcm-12-00755-t001:** Demographics and peri-operative characteristics of the study cohort (No. = 251).

Age (Years)	
Median (IQR)	14 (13–16)
Range	9–21
BMI (kg/m^2^)	
Median (IQR)	20.1 (17.9–21.6)
Range	14.4–26.1
Varicocele clinical grade (No. (%))	
III	183 (73%)
II	9 (4%)
Type of presentation (No. (%))	
Primary	235 (94%)
Recurrent after open surgery	16 (6%)
Operative time (min)	
Median (IQR)	35 (30–45)
Range	12–80
Presence of collateral veins (No. (%))	114 (45.4%)
Type of SA (No. (%))	
Aethoxysclerol	117 (46.6%)
Fibrovein	134 (53.4%)
Amount of SA (mL)	
Median (IQR)	3 (3–4)
Range	1.5–8
Fluoroscopy time (min)	
Median (IQR)	38.5 (27.7–54)
Range	16–149
DAP (cGycm^2^)	
Median (IQR)	89.6 (62.5–143.9)
Range	22.9–432.9
Effective Dose (mSv)	
Median (IQR)	0.19 (0.14–0.31)
Range	0.06–0.73

Keys: BMI = body mass index; SA= sclerosing agent; DAP = dose area product.

**Table 2 jcm-12-00755-t002:** Postoperative characteristics of the study cohort (No. = 251).

Duration of Follow-Up (Months)	
Median (IQR)	6 (3–12)
Range	
Clinical recurrence (No. (%))	5 (2.1%)
Complications (No. (%))	2 (0.81%)
Failed procedure (No. (%))	1 (0.4%)

**Table 3 jcm-12-00755-t003:** Linear regression models predicting radiation exposure in the whole cohort (No. = 251).

	UVA Model Beta; *p*-Value (95% CI)	MVA Model Beta; *p*-Value (95% CI)
Age	5.72; 0.016 (1.09–10.35)	1.18; 0.732; (−5.693–8.053)
BMI	11.026; <0.001 (5.39–16.66)	12.96; <0.001 (5.97–19.95)
Collateral veins	14.36; 0.193 (−7.3–36.03)	
>3 mL SA used	32.97; 0.003 (11.10–54.83)	15.204; 0.369 (−18.38–48.792)
Operative time	1.46; 0.012 (0.32–2.59)	1.99; 0.017 (0.367–3.62)

Keys: BMI, body mass index; CI, confidence interval; MVA, multivariate model; UVA, univariate model.

## Data Availability

Data available on request due to restrictions.

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
