# Peer review of "Radiation Exposure and Surgical Outcomes after Antegrade Sclerotherapy for the Treatment of Varicocele in the Paediatric Population: A Single Centre Experience"

_jcm, 2023, doi:10.3390/jcm12030755_

Round 1

Reviewer 1 Report

This study is a single-center experience, which proved that Tauber procedure is safe for the treatment of varicocele in the Pediatric population, and also it associated with low effective dose. The work done in this submission is sound and good science. Methods are adequately described, there are no obvious errors in the statistics. Conclusions are supported by the results. 

Author Response

The authors thank the reviewer for this comment. Varicocele is one of the treatable causes of male infertility and every effort should be made towards finding the best possible treatments and most appropriate indications.

Reviewer 2 Report

well written study. This sclerosing agents can cause serious complication, Even pulmonary embolism can be seen, especially in applications to veins. Authors should improve the discussion section about more complications of this procedure. 

Especially in the adolescent age group, radiation can be a great risk. It should not be forgotten that the place of this treatment in Urology guidelines is controversial. Discussion on this issue should also be expanded.

Author Response

Thank you very much for your review and valuable insight. The discussion section of our manuscript has been revised and now mentions possible complications related to the use of sclerosing agents in addition to a detailed examination of the controversial role of sclerotherapy for varicocele treatment in the 2022 EAU pediatric urology guidelines.

Although rare cases of thrombus migration and pulmonary embolism are reported in the literature after varicocele embolization, we did not incur in any major thromboembolic event in our cohort.

For what concerns radiation safety, the authors believe that every effort should be made to recognise radiation risk, to promote radiation safety, and to optimize dose reduction techniques, especially when treating young males that are affected by a benign pathology.

The main aim of our study is to provide reasonable estimates of radiation exposure during Tauber surgery as data in the literature is very scarce and outdated in this regard. To the best of our knowledge no study provides a detailed report of measures and estimates of radiation exposure itself during either antegrade or retrograde sclerotherapy in the pediatric population.

As a matter of fact, the 2022 EAU pediatric urology guidelines state that angiographic occlusion of the spermatic veins meets the requirements for lymphatic sparing repair, which has a strong recommendation to prevent hydrocele formation and testicular hypertrophy. However, the same guidelines criticise fluoroscopy guided techniques for their radiation burden, stating that the radiation exposure is less controllable with the antegrade approach. Noticeably, this statement is based on dated studies examining small cohort of patients and providing little details on the radiation burden.  Thon et al, in 1989, presented one the first experiences of percutaneous sclerotherapy for varicocele treatment in 31 children of 10 to 16 years of age, affirming that fluoroscopic radiation exposure could last as long as 14 minutes (mean 4.9 minutes). Fifteen years later, Wunsch et al examined a very large cohort of pediatric patients treated with retrograde sclerotherapy, the vast majority of which were treated from 1981 to1996. The authors stated that the radiation dose depended on the duration of the probing of the internal spermatic vein with doses area product measurements at 50-150 cGy*cm2, by using appropriate apertures, although a statistical analysis was not provided. In 2010, also Fayad et al. described their experience with percutaneous retrograde endovascular occlusion (PREVO) in 71 pediatric varicocele patients, treated with high success rates and low complication rates between the years 1990 and 2004. However, no data was provided on the radiation burden of the procedure. To conclude, further studies providing a detailed analysis of radiation exposure are needed in order to confirm our findings, as currently available literature does not provide adequate information on this matter. 

Round 2

Reviewer 2 Report

none

Author Response

The authors thank the Academic Editor for this note.

  1. The text has been revised accordingly (line 152-153)
  2. Line 59: the authors agree with the Academic Editor and the text has been revised accordingly.
  3. Line 101-102: This statement is an oversight and the authors agree with the Academic Editor that the wording of the sentence is confusing. In our technique the scrotum is opened as described by the Editor. As a matter of fact, the only layer of the tunica vaginalis that may be opened when searching for the pampiniform plexus veins is the lamina parietalis of the tunica vaginalis, which can sometimes extend upwards in front and on the medial side of the cord. However, as the editor pointed out, the tunica vaginalis, that covers the testis and epididymis, is not opened. Hence, the text has been revised to “After opening the scrotum, the pampiniform plexus veins are identified and the most dilated and straight vessel is selected, isolated and prepared for cannulation.”
  4. Line 169: The text has been revised accordingly

Sincerely yours,

Carolina Bebi on behalf of all the authors
